# Genomic and Temporal Analysis of Deletions Correlated to qRT-PCR Dropout in N Gene in Alpha, Delta and Omicron Variants

**DOI:** 10.3390/v15081630

**Published:** 2023-07-26

**Authors:** Giulia Gatti, Martina Brandolini, Andrea Mancini, Francesca Taddei, Silvia Zannoli, Giorgio Dirani, Martina Manera, Valentina Arfilli, Agnese Denicolò, Anna Marzucco, Maria Sofia Montanari, Irene Zaghi, Massimiliano Guerra, Rita Tennina, Maria Michela Marino, Laura Grumiro, Monica Cricca, Vittorio Sambri

**Affiliations:** 1Department of Medical and Surgical Sciences (DIMEC)—Alma Mater Studiorum, University of Bologna, 40138 Bologna, Italy; giulia.gatti12@unibo.it (G.G.); martina.brandolini@outlook.it (M.B.); monica.cricca3@unibo.it (M.C.); 2Unit of Microbiology, The Greater Romagna Area Hub Laboratory, 47522 Cesena, Italy; andrea.mancini@auslromagna.it (A.M.); fra.taddei@hotmail.it (F.T.); silvia.zannoli@auslromagna.it (S.Z.); giorgio.dirani@auslromagna.it (G.D.); martina.manera@auslromagna.it (M.M.); valentina.arfilli@auslromagna.it (V.A.); agnese.denicolo@auslromagna.it (A.D.); anna.marzucco@auslromagna.it (A.M.); sofi.monta.msm@gmail.com (M.S.M.); irene.zaghi@auslromagna.it (I.Z.); massimiliano.guerra@auslromagna.it (M.G.); laura.grumiro@auslromagna.it (L.G.); 3Unit of Laboratory Medicine—Local Health Authority 1 Complex Operative Unit, 67051 L’Aquila, Italy

**Keywords:** WG-NGS, SARS-CoV-2, deletions, gene dropout, *N* gene

## Abstract

Since the first SARS-CoV-2 outbreak, mutations such as single nucleotide polymorphisms (SNPs) and insertion/deletions (INDELs) have changed and characterized the viral genome sequence, structure and protein folding leading to the onset of new variants. The presence of those alterations challenges not only the clinical field but also the diagnostic demand due to failures in gene detection or incompleteness of polymerase chain reaction (PCR) results. In particular, the analysis of understudied genes such as *N* and the investigation through whole-genome next generation sequencing (WG-NGS) of regions more prone to mutate can help in the identification of new or reacquired mutations, with the aim of designing robust and long-lasting primers. In 48 samples of SARS-CoV-2 (including Alpha, Delta and Omicron variants), a lack of *N* gene amplification was observed in the genomes analyzed through WG-NGS. Three gene regions were detected hosting the highest number of SNPs and INDELs. In several cases, the latter can interfere deeply with both the sensitivity of diagnostic methodologies and the final protein folding. The monitoring over time of the viral evolution and the reacquisition among different variants of the same mutations or different alterations within the same genomic positions can be relevant to avoid unnecessary consumption of resources.

## 1. Introduction

As consequence of an increment of the worldwide infection cases, on 11 March 2020, the World Health Organization (WHO) declared the coronavirus disease 2019 (COVID-19) a global emergency [1]. The infectious agent was identified in a new viral species classified in the *Coronaviridae* family and named Severe Acute Respiratory Syndrome Coronavirus 2 (SARS-CoV-2), phylogenetically related to beta-coronaviruses SARS-CoV and MERS-CoV [1,2]. 

Once the viral genome was sequenced, SARS-CoV-2 has been considered genetically similar to SARS-CoV, notwithstanding the major genomic modifications that differentiate the two that occur in the spike gene (S). The gene product, the S protein present on the surface of the virion, mediates the attachment to human angiotensin-converting enzyme 2 (ACE2) by the receptor binding domain (RBD). This region is highly variable, hence it has been reported that the insertion of six amino acid mutations in the RBD of SARS-CoV-2 facilitates the ligation of the viral receptor to the enzyme [2,3,4].

Equally, the acquisition of 12 nucleotides increases the spike protein’s overall instability and facilitates the furin cleavage as well as the viral infectivity by encoding the amino acid sequence PRRA, HRRA or LRRA at the level of the junction between S1 and S2 subunits [2,3,4,5]. In this scenario, COVID-19 has caused millions of deaths and due to its continuous circulation, the viral genome mutated and originated new variants that may be characterized by resistance to medical measures [6]. The new arising strains have been named through the Greek alphabet and classified into variants of concern (VOCs), variants of interest (VOIs) and under monitoring (VUMs) depending on their frequency, infectiveness and mortality [7]. To respond to the need for a validated and standardized detection, according to the WHO, a robust workflow for the identification of SARS-CoV-2 has to include nucleic acid amplification testing (NAAT) such as reverse-transcription polymerase chain reaction (qRT-PCR) on E, RdRP, *N* and *S* genes [8]. But some of the mutations acquired during the evolution of the viral genome may lead to failure in RT-PCR amplification, for instance the 69-70 del in the S gene [9,10] as well as mutations occurring in the N gene [11,12]. Globally, this technique is considered the gold standard for the diagnosis of COVID-19 due to its sensitivity; therefore, many industries had developed commercial kits to reduce the typical run time and improve the methodology protocol [13,14,15]. The importance of a solid and proven stability of genomic regions is pivotal to design targets for a diagnostic kit, therefore in this study we compared 48 samples identified during the daily diagnostic workflow presenting a PCR peculiar result. The reference assay used was the Allplex SARS-CoV-2 Extraction-Free (Seegene Inc., Seoul, Republic of Korea) with which the samples exhibited a lack of amplification of the *N* gene. Samples were classified into Alpha, Delta or Omicron after whole-genome next generation sequencing (WG-NGS) on the Illumina platform and retested for confirmation on Xpert Xpress SARS-CoV-2 (Cepheid, Sunnyvale, CA, USA) assay technology.

## 2. Materials and Methods

### 2.1. Samples Collection

The study was performed on 48 clinical nasopharyngeal samples from SARS-CoV-2 infected individuals collected during the daily routine workflow from March 2021 to July 2022. All the samples used in this study underwent the anonymization procedure used at the Unit of Microbiology of the Hub Laboratory of the Great Romagna Area (Pievesestina FC, Italy) to adhere to the regulations issued by the local ethical board (AVR-PPC P09, rev.2; based on Burnett et al., 2007). The study was conducted according to the guidelines of the Declaration of Helsinki and approved by the Institutional Review Board of AUSL Romagna (protocol code “COVdPCR” of 7 February 2020).

### 2.2. RT-PCR Amplification

The presence of SARS-CoV-2 was evaluated through the Allplex SARS-CoV-2 Extraction-Free system, a multiplex qRT-PCR based on TaqMan probes targeting four viral genes: *E*, RdRP/*S* and *N*. The Extraction-Free Assay relies on a dilution of 15 µL of sample in 45 µL of nuclease-free water and an incubation at 98 °C for 3 min. Then the sample is maintained at 4 °C for 5 min. Successively, master mix preparation and reaction setup were conducted as described in the manufacturerinstructions. A total volume of 15 µL of master mix was aliquoted in a tube containing 5 µL of extracted sample [16]. After the PCR-setup step, the plate was transferred to a CFX96 thermocycler (Bio-rad, Hercules, CA, USA) where results analysis and target quantification were performed with Seegene Viewer software from Seegene Inc. The positivity was declared in samples presenting Cycle Threshold (Ct) values below the manufacturer’s suggested ones of 40. When an *N* gene dropout was detected on the Allplex SARS-CoV-2 Extraction-Free system, the sample was retested through the Xpert Xpress SARS-CoV-2 for GeneXpert Dx or GeneXpert Infinity systems [17]. The technology is based on the use of a single cartridge containing reagents and components for a single RT-PCR on a 300 µL of sample. Each cartridge includes a sample processing control (SPC) to ensure an appropriate reaction amplification in temperature, time and reagents. Additionally, a probe check control (PCC) validates the fluorescence stability. The technique targets two SARS-CoV-2 genes that are *E* and *N*2.

### 2.3. Viral RNA Extraction and Next-Generation Sequencing

The genetic material was isolated and clarified through automated extraction and purifications using the Maelstrom 9600 system (TANBead—Taiwan Advanced Nanotech Inc., Taiwan). The sample undergoes a mechanical lysis through magnetic beads and three washing steps. The extracted RNA is eluted in about 50 µL.

After the extraction, the library preparation was performed following the CleanPlex SARS-CoV-2 Flex Research and Surveillance NGS Panel (Paragon Genomics, Inc., Hayward, CA, USA). According to the manufacturer’s instructions, protocol steps of reverse transcription of viral RNA, digestion, indexing PCR and purifications were performed on an automated workstation system distributed by Hamilton (Reno, NV, USA). Once the library was generated, the cDNA was quantified, serial diluted and sequenced on the MiSeq platform (Illumina, San Diego, CA, USA) [18].

### 2.4. Data Analysis

Forward and Reverse FastQ files were obtained after sequencing, and reads trimmed and aligned on the Wuhan reference genome (NCBI Accession number: NC_045512.2) using SOPHiA DDM software (SOPHiA Genetics^TM^, Lausanne, Switzerland). The overall reads length was maintained within the range of 149–151. To avoid spurious alignments, a high depth of coverage (DC) was calculated: a possible deletion was manually inspected and included in the study when the DC was around 100 mapped reads.

The threshold for the variant calling (VF) was set at 70% of the total number of mapped reads in a specific genome position. Then, the associated consensus sequence was generated in a FASTA file for each sample and the percentages in presence of each mutation were compared. The classification and categorization of SARS-CoV-2 variants were performed on the PANGO Lineages [19,20] website (v.4.2) and confirmed on the Nextclade online web application (v. 2.13.0) by Nextstrain [21,22].

Deletions were manually inspected by using the desktop application of Integrative Genomic Viewers (IGV) [23].

Raw data have been deposited in the Sequence Read Archive (SRA), BioProject accession number PRJNA970221.

### 2.5. Reference Consensus Sequences Generation

To define the general nationwide mutational trend of each variant, three local consensus sequences (LCS) were generated for 50 random Alpha sequences, 48 Delta and 43 Omicron variants according to the lowest percentage of 1 of not assembled nucleotides (%N) and excluding the testing samples. The random sequences were chosen within a limited temporal range starting from the earliest detection of the analyzed variant up to one month, to annotate the original genome of strain arrived in Emilia-Romagna and perform the identification of putative mutations cause of the gene dropout. The final LCSs were annotated in the FASTA file on Lasergene MegAlign Pro software (DNASTAR, Inc., Madison, WI, USA) and, according to the previous classification performed on Pangolin, the tested genomes were aligned to the reference consensus sequence of the variant they belong to. The differences in mutational presence and distribution were investigated.

Successively, sequences were aligned together with the reference SARS-CoV-2 genome through ClustalOmega [24] and the remaining mutations on the *N* gene were analyzed.

### 2.6. N Protein Structure Prediction

The nucleotide sequences were translated into the corresponding amino acids chain through the Transeq EMBOSS tool [25] and then aligned to the NCBI reference N protein (YP_009825061.1). The prediction of the 3D structure of the N protein was calculated through RoseTTAFold [26], the algorithm was set with the default parameters and 5 probabilistic models were generated associated with an Angstrom error rate estimation for every single amino acid [27]. Then, the proteins were analyzed on Lasergene Protean 3D software (DNASTAR, Inc., Madison, WI, USA).

## 3. Results

### 3.1. Ct Values

Once the PCR results for both the technologies used in this study were available, data were compared to confirm the *N* gene dropout found for the Seegene’s assay (Table 1).

### 3.2. Sequence Analysis

The analyzed samples were classified into 31 Alpha (B.1.1.7), 10 Delta (sublineages AY.23, AY.43 and AY.102) and 7 Omicron (sublineages BA.1.1, BA.1.21, BA.2, BA.2.36, BA.2.12.1 and BA.5). The *N* gene was investigated, and three regions were identified where mutations were present in least 1 out 48 samples. The *N* gene sequence considered starts at 28,274 nucleotide position of the reference genome. The first identified region 1 (R1) covers nucleotides from positions 28,280 to 28,703; region 2 (R2) extends for 27 nucleotides from 28,879 to 28,907 and has been identified as the shortest zone. Region 3 (R3) includes genomic positions from 28,912 to 29,510 (Figure 1).

The DC and VF were manually inspected and visualized for each detected mutation, to discriminate if a deletion could be included in the study (Appendix A). Then, after mapping the mutations, the regions were analyzed following the SARS-CoV-2 evolution and temporal onset of variants. Particularly, a deletion of six nucleotides was found in position 28,890–28,895 in the R2 of 23 Alpha samples; the mutation resulted in a deletion of two amino acids, alanine and proline (Pro207_Ala208del). Within the same region, a deletion of three nucleotides was annotated in seven Alpha samples and two Omicron (BA.1 and BA.1.21) in positions from 28,896 to 28,898. The deletion was called Ala208_Arg209del, where the sequence GCTAGA was changed in a glycine (GGA). The sole exception among Alpha samples was the number 11, which did not present any of the previous mutations but a deletion named Arg203_Ser206del in nucleotide positions 28,879–28,890.

Alongside the deletions, one SNP was annotated at nucleotide 28,881 in all three variants: G > A in Alpha and Omicron and G > T in Delta. Additionally, the two registered SNPs were G > A and G > C in Alpha and Omicron variants at nucleotide positions 28,882 and 28,883, respectively. 

In the remaining samples that did not present these mentioned mutations, some other deletions were annotated. 

Particularly, in two Omicron variant samples (46 and 48), respectively, BA.2.38 and BA.5.1, two mutations were found in R2 alongside the Glu31_Ser33del starting from 28,881 (Gly203_Ala208del with threonine insertion) and ending at 28,895 as well as a deletion from 28,898 (Arg209_Met210del) ending at 28,903. Similarly, a deletion from 28,899 to 28,907 (Arg209_Gly212del) was found in sample 45 (BA.2) that resulted in a serine insertion. Lastly, a long deletion of 18 nucleotides (Gly204_Arg209del) was acquired in the sequence of sample 42 (BA.1.1). 

A similar mutational event was found in the R3 of 10 Delta samples and 1 BA.5.1 sample that presented a deletion of six nucleotides from 28,912 to 28,917; the mutation resulted in the omission of glycine 214 and 215 (Gly214_Gly215del). Moreover, at position 29,510, four Omicron sequences had an SNP in A > C. 

Even in R1 of seven Omicron samples, a deletion in positions from 28,362 to 28,370 was annotated and called Glu31_Ser33del. Concerning SNPs of the Alpha variant, the R1 presented mutations at 28,280 (G > C), 28,281 (A > T) and 28,282 (T > A) genomic positions. At nucleotide 28,299, five Delta mutated A > T and all Delta sequences changed A > G at nucleotide 28,461. On the contrary, a reversion A > T can be seen at position 28,311 of Omicron variants. 

Deletions were also visualized in Integrative Genomic Viewers (IGV) (Appendix A).

### 3.3. LCS Comparison

Alpha: The generated LCS presented three substitutions at nucleotide positions 28,280 (G > C), 28,281 (A > T) and 28,282 (T > A), originating the amino acidic mutation D3L. Three SNPs were found at positions 28,881 (G > A), 28,882 (G > A) and 28,883 (G > C) that resulted in lysine and arginine mutation. The last SNP, C > T, was annotated in the sequence at 28,977 and was associated with the substitution called S235F.

Delta: In the Delta LCS sequence, the *N* gene presented four mutations. The SNP A > G at 28,461 resulted in the D63G mutation, alongside the amino acid substitutions G > T at 28,881 (R203M) and G > T at 28,916 (G215C). At genome position 29,402, the SNP G > T generated the amino acidic substitution D377Y.

Omicron: the genome of the Omicron LCS was investigated, and four mutations were found to be present in the *N* gene. At position 28,311, the C > T mutation originated the amino acid substitution P13L, and three SNPs at positions 28,881 (G > A), 28,882 (G > A) and 28,883 (G > C) resulted in lysine and arginine mutations, as occurred in Alpha LCS.

Once the mutational set of the three LCSs was characterized, the annotated SNPs were not considered because of the dropout when present in sample sequences. By excluding those mutations, some exceptions were noted as sporadic SNPs among samples sequences, although the main divergence was the presence of the deletion Gly214_Gly215 in place of mutation G > T at nucleotide 28,916 in Delta sample sequences (Appendix A).

### 3.4. Temporal Analysis

Due to the recurrence of some deletions in different patients, the epidemiological distribution and the onset of mutations were investigated through an internal database which counted the total number of sequences acquired during the sequencing surveillance of the local COVID-19 cases. The Pro207_Ala208del was thereby annotated in Emilia-Romagna in 25 Alpha sequences from late March to late May 2021. Around the same period, the Ala208_Arg209del appeared in three Alpha samples in early April and remained until early June 2021 for a total of twelve samples; notably, the same mutation was found again in two Omicron sequences in February and May 2022. From November to December 2021, the deletion Gly214_Gly215 characterized the sequence of 11 patients, all showing dropout in the *N* gene. The Glu31_Ser33del appeared to be part of the Omicron variant mutational arrangement in late December 2021 and constantly increased its fixation until it became constitutive; therefore, it cannot be directly responsible for the lack of amplification. On the other side, Arg203_Ser206del in one Alpha sample and Gly203_Ala208del, Gly204_Arg209del, Arg209_Met210del and Arg209_Gly212del present in four Omicron samples were unique in the whole database (Figure 2).

### 3.5. D Structure Prediction

The structural coordinates of three of the most common deletions (Pro207_208del, Ala208_209del and Gly214_215del) in the protein folding were analyzed. The mutations Pro207_208del and Ala208_209del were present in a link zone between two flexible positive-charged regions. Regarding the deletion Gly214_Gly215, the area was a flexible zone between a turn region and an α-structure. The Omicron characteristic deletion of Glu31_Ser33 affected a positive-charged flexible turn area (Figure 2).

## 4. Discussion

The spreading of SARS-CoV-2 infections and the following emergence of variants led not only to enhanced fitness and virulence but also to a decrease in the accuracy of diagnostic RT-PCR kits or, specifically, in the efficiency of primers and probes consequent to the onset of mutations and deletions [28,29]. Despite the continuous effort for a rapid development of RT-PCR-based techniques, some alterations may destabilize the hybridization of primers and result in a gene dropout or a delay in Ct values. In particular, the evolution of the N gene as part of diagnostic targets requires further in-depth study built around the necessity of strengthening the diagnostic robustness of commercial methodologies [30]. 

On account of this, the whole-genome sequencing surveillance of COVID-19 allowed the characterization and classification of mutations and variants [31] and its exploitation gains increasing relevance in the prediction of the geographical distribution and phylogenetic evolution of the virus as well as a blueprint in the advancement of molecular diagnostic assays [29]. 

In this study, the strong connection between the temporal analysis and the genomic surveillance of SARS-CoV-2 has been crucial for the identification of three regions of the *N* gene prone to mutate and that present large deletions that may lead to a dropout in *N* gene amplification curve. Specifically, Glu31_Ser33del found in R1 of Omicron variants is not chargeable of dropout because the frequency of the mutation has exponentially increased with the enlargement of viral diffusion, and the deletion has been fixed during the evolution of the virus as to be included within the list of characteristic mutations of lineage B.1.1.529 and its subclassifications [20]. The mutation was not included in the Omicron LCS; therefore, authors assume that the deletion was not present at the beginning of the variant spreading in Emilia-Romagna. To exclude the dropout causative effect of the Glu31_Ser33del, mutations in R2 and R3 of Omicron samples were considered: all sequences presented at least a deletion in one of the two regions. Particularly, all Omicron sequences except for sample 48 acquired the Ala208_Arg209del that is shared with Alpha; on the other hand, the Gly214_Gly215del in common with Delta samples is found in sample number 48. 

Therefore, in a situation opposite to that of R1, deletions occurring in R2 or R3 may have a causative relation to the lack of *N* gene amplification using the Allplex SARS-CoV-2 Extraction-Free Assay by Seegene. In fact, the Gly214_215del found in Delta and Omicron variants was also detected in other studies and associated with a failure in qRT-PCR results [32,33]. Previously, the Gly214_215del was only found in the AY.4 sublineage, but in our study the deletion is annotated not only in additional and different AY strains but also in an Omicron sequence that showed the same *N* gene dropout as well as in Delta. Therefore, the deletion may be reacquired over time and viral evolution. Hence the monitoring of the R3 region, with a particular focus on the positions from 28,913 to 28,918 nucleotides, can be proposed to avoid probable PCR failures in new commercial kit development. Additional uncommon mutations such as deletions Ala208_Arg209del, Pro207_Ala208del and Arg203_Ser206del were observed associated with a *N* gene dropout. 

According to the study of Laine et al., the Ala208_Arg209del was found in Finland associated with false-negative *N* gene results [34].

More broadly, the gene area covered by R2 and R3 belongs to a link region rich in serine and arginine whose functions relate directly to the capacity of RNA separation and solubilization. Given that the nucleocapsid is one of the regulators of SARS-CoV-2 RNA transcription, mutations or deletions occurring in those areas may have an impact also on the viral fitness, replication and host interaction [35,36,37].

## 5. Conclusions

The *N* gene continues to be an understudied gene; therefore, tracking over time of its mutational trend remains decisive for the identification of more stable regions or positions that may represent fundamental acquisitions for industries interested in commercial kits development and differential diagnosis.

By carefully analyzing the *N* gene sequence, some regions can be identified that are more stable than others. With this in mind, designing primers for stable regions and not for areas prone to mutate can prevent the manufacturing company from upgrading and improving assay even after the enactment of the commercial kit. Likewise, the user could achieve greater test reliability by reducing false-negative results.

## Figures and Tables

**Figure 1 viruses-15-01630-f001:**
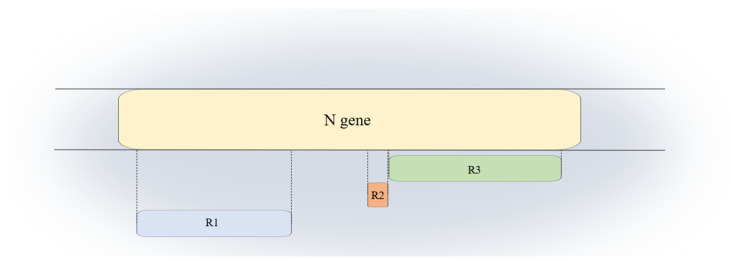
The extension and the position of the three regions were mapped on the *N* gene. Region 1 (R1) extends from nucleotides 28,280 to 28,703; region 2 (R2) resulted in the shortest among the three and includes nucleotides from 28,879 to 28,907; the third region (R3) lies at the end of the gene, from nucleotides 28,912 to 29,510.

**Figure 2 viruses-15-01630-f002:**
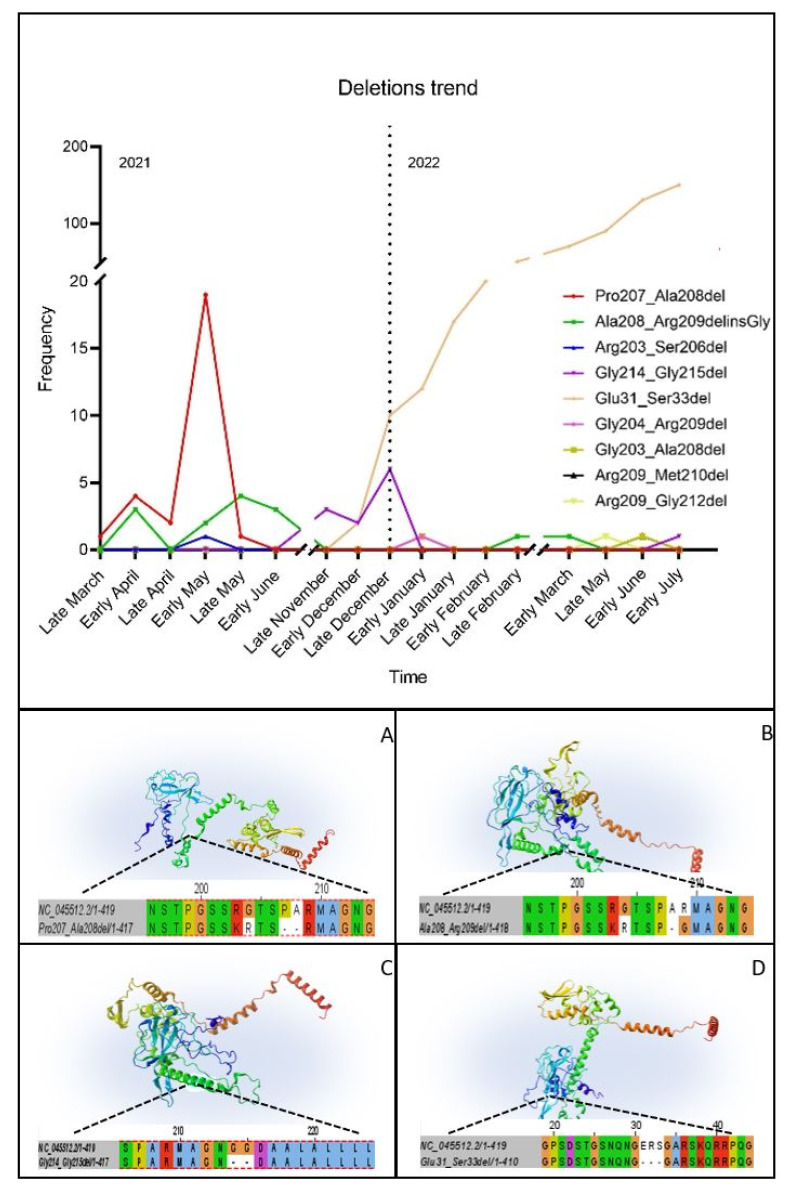
The mutation frequency was traced over time and registered for over one year. According to their trend, the 3D structures of N proteins presenting the most recurrent deletions were generated through RoseTTAFold, and images were created using Lasergene Protean 3D software from DNASTAR, Inc. The protein sequences were aligned to the Wuhan reference YP_009825061.1. The four deletions occurred in flexible link regions: (**A**) Pro207_Ala208del; (**B**) Ala208_Arg209del with a Glycine insertion; (**C**) Gly214_Gly215del; and (**D**) Glu31_Ser33del.

**Table 1 viruses-15-01630-t001:** The Ct values obtained from the two methodologies were compared to confirm the lack of amplification of the *N* gene. The samples tested with the Allplex SARS-CoV-2 Extraction-Free Assay were confirmed positive with the Xpert Xpress SARS-CoV-2 Assay through which the amplification of the *N*2 target was present.

Sample Number	Variant	Allplex SARS-CoV-2 Extraction-Free Assay				Xpert Xpress SARS-CoV-2 Assay		
		*E* Gene	RdRP/*S* Gene	*N* Gene	IC *	E Gene	*N*2 Gene	IC *
1	B.1.1.7	25.98	29.02	N/A	22.05	20.9	24.1	27.6
2	B.1.1.7	21.23	22.34	N/A	22.45	17.1	17.9	27.5
3	B.1.1.7	19.18	21.25	N/A	22.12	14.9	16.3	27.5
4	B.1.1.7	19.12	22.01	N/A	21.98	15.8	17.1	27.1
5	B.1.1.7	24.45	26.68	N/A	22.01	20	21.7	27.2
6	B.1.1.7	19.04	21.32	N/A	21.77	15	16.6	28.3
7	B.1.1.7	23.12	24.23	N/A	22.13	18.6	19.3	27.4
8	B.1.1.7	25.98	28.1	N/A	22.31	21.8	23.4	27.7
9	B.1.1.7	27.98	30.01	N/A	21.89	22.9	25.4	28.1
10	B.1.1.7	26.78	29.07	N/A	22.52	21.8	24.7	28.5
11	B.1.1.7	24.98	28.13	N/A	22.33	20.7	23.3	28.2
12	B.1.1.7	23.37	23.98	N/A	21.88	19.7	19.8	27.3
13	B.1.1.7	23.01	26.25	N/A	21.91	18.5	21.4	27.1
14	B.1.1.7	24.55	27.32	N/A	22.12	20.2	23.1	28.3
15	B.1.1.7	23.72	26.03	N/A	22.37	18.7	21.3	27.5
16	B.1.1.7	20.34	22.22	N/A	22.41	16.12	17.6	28.0
17	B.1.1.7	21.23	24.37	N/A	22.4	17.06	19.5	28.1
18	B.1.1.7	18.11	21.01	N/A	21.83	14.94	16.3	27.9
19	B.1.1.7	19.23	19.99	N/A	21.85	14.78	15.3	27.4
20	B.1.1.7	22.12	23.92	N/A	22.01	17.63	19.5	28.0
21	B.1.1.7	19.15	22.32	N/A	21.99	15.06	17.8	28.4
22	B.1.1.7	23.04	24.59	N/A	22.38	19.06	20.9	27.8
23	B.1.1.7	24.46	27.12	N/A	22.52	20.1	23.0	27.1
24	B.1.1.7	20.67	21.89	N/A	22.04	16.4	17.7	27.3
25	B.1.1.7	28.47	30.15	N/A	22.56	24.1	25.6	28.2
26	B.1.1.7	25.34	29.07	N/A	22.63	21.1	24.3	28.5
27	B.1.1.7	23.34	25.87	N/A	22.43	19.5	22.1	27.3
28	B.1.1.7	26.76	29.1	N/A	22.76	22.3	24.8	27.8
29	B.1.1.7	23.53	26.67	N/A	22.51	19.7	22.4	27.3
30	B.1.1.7	24.79	27.2	N/A	22.49	21.1	23.0	28.1
31	B.1.1.7	19.55	21.68	N/A	21.81	15.2	17.5	27.7
32	AY.102	22.13	23.54	N/A	22.06	18.1	19.1	27.4
33	B.1.617.2	17.77	21.98	N/A	21.86	14.8	18.2	28.5
34	AY.43	23.51	25.12	N/A	22.32	19.6	21.0	28.5
35	AY.43	21.88	24.56	N/A	22.44	18.2	20.2	27.0
36	AY.43	25.34	27.89	N/A	22.61	21.8	24.1	27.4
37	AY.43	18.87	21.03	N/A	21.93	15	16.6	28.6
38	AY.23	18.56	20.45	N/A	22.14	15.2	17.0	27.9
39	AY.23	23.34	25.12	N/A	22.25	19.3	20.6	28.6
40	AY.23	20.06	22.76	N/A	22.17	16	18.2	27.2
41	AY.23	26.34	28.98	N/A	22.78	22.1	25.0	27.6
42	BA.1.1	17.88	19.34	N/A	22.21	14.4	15.4	28.1
43	BA.1	20.91	22.65	N/A	22.01	16.5	18.4	27.3
44	BA.1.21	20.67	21.76	N/A	22.32	17.1	18.2	27.5
45	BA.2	17.76	18.63	N/A	21.97	14.2	14.0	27.2
46	BA.2.36	20.09	22.87	N/A	22.31	16.1	18.0	28.0
47	BA.5	29.03	29.77	N/A	22.4	24.7	25.7	28.3
48	BA.5.1	18.36	20.87	N/A	22.35	15.1	16.0	27.9

* IC: internal control.

## Data Availability

The data presented in this study are available on request from the corresponding author. The data are not publicly available due to privacy.

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
