# Peer review of "Genomic and Temporal Analysis of Deletions Correlated to qRT-PCR Dropout in N Gene in Alpha, Delta and Omicron Variants"

_viruses, 2023, doi:10.3390/v15081630_

Round 1
Reviewer 1 Report (Previous Reviewer 1)
The brief report work presented by Giulia Gatti and colleagues titled “Genomic and temporal analysis of deletions correlated to qRT-PCR dropout in N gene in Alpha, Delta and Omicron variants” is well written, clear, and easy to read. The topic is very interesting and therefore, it adds clustered information to the subject area of infectious diseases mediated by SARS-CoV2 infection. It is still a cutting edge area and we still not have drug against the COVID-19 disease. In particular, the author performed a very well-conceived overview about the role of N viral protein its structure and function and change during evolution of variants of concern. Other proteins change during virus evolution rapidly and it is the case to mention that in the discussion section.
Please consider these references https://doi.org/10.3390/reports5020014; https://doi.org/10.1128/jcm.03278-20; https://doi.org/10.3390/biomedicines10081839; 10.1016/j.virusres.2021.198398
Author Response
Dear Reviewer 1,
the authors thank you for your revision and comment. We added the references that you suggested:
- Line 64: citation number 10: http://doi.org/
10.3390/reports5020014 - Line 65: citation number 11 (https://doi.org/10.1016/j.virusres.2021.198398)
- Line 314: citation number 36 (https://doi.org/
10.3390/biomedicines10081839) - Line 314: citation number 37 (https://doi.org/10
.1128/JCM.03278-20.)
Line 334: the authors added in the Funding section "The PhD scholarship of Giulia Gatti was funded by the European Union - NextGenerationEU through the Italian Ministry of University and Research under PNRR – Mission 4 Component 2, Investment 3.3 “Partnerships extended to universities, research centres, companies and funding of basic research projects” D.M. 352/2021 – CUP J33C22001330009."

Reviewer 2 Report (New Reviewer)
The authors investigated samples of Alpha, Delta and Omicron variants, in which amplification of N gene in qRT-PCR failed. The purpose of this study was identification of new or reacquired mutations with the aim to design robust and long-lasting primers. I believe that this purpose was achieved.
Author Response
Dear Reviewer 2,
the authors thank you sincerely for your revision and comment. In the hope to make an improvement in the scientific knowledge, we send our warmest greetings.
This manuscript is a resubmission of an earlier submission. The following is a list of the peer review reports and author responses from that submission.
Round 1
Reviewer 1 Report
The review work presented by Giulia Gatti and colleagues titled “Genomic and temporal analysis of deletions correlated to qRT-PCR dropout in N gene in Alpha, Delta and Omicron variants” and colleagues is well written, clear, and easy to read. The topic is interesting and therefore, it adds clustered information to the subject area of covid pandemic/epidemic, which is still a cutting-edge area. In particular, the author performed a very well-conceived brief report.
Minor
Please clarify putting more details in the “RT-PCR amplification” section of each method used, especially the Ct number of the total run.
The drop-off could be also due to the different complementarity of the primers’ probe, crucial for PCR, where mismatch can give up to 7 Ct of difference. For reference 10.2353/jmoldx.2010.090035; 10.3390/reports5020014.
Reviewer 2 Report
The paper by Gatti et al attempts to explain the qRT-2 PCR dropout in the N gene. The text is quite short and very unclearly written. It will greatly benefit from some input of native English speakers, as English is a really weak spot here. Clarity of thought suffers as a result; novelty appears very questionable so much more work is needed to make it into a potentially acceptable paper. The use of the word “nCoV” is alarming – the world stopped using this word 3 years ago, do the authors know that?
The very first sentence of Abstract sets a tone of rather inept English. "Diffusion" is misused here, but the rest is alarming:
"mutations as Single Nucleotide Polymorphisms (SNPs) and Insertion/Deletion (INDELs) have elicited and characterized modifications in genome sequence".
Well, mutations are these modifications, how can they "elicit and characterize modifications"? It's like saying "people in Italy elicit and characterize people in Italy".
The rest "enough for the onset of new viral variants" is even worse. I also wonder what is not enough?
Thus, the whole Abstract (and most of the text) must be rewritten for clarity, avoiding such strange and basically meaningless phrases.
Some more comments on grammatical and other English issues to help improve the text (these are very incomplete, just to give the authors an idea of what type of mistakes to correct), line numbers given:
41 pretty bad English, obscuring the meaning
48-49 strange use of hyphens plus things like “resulting encoding” and “along with viral infectivity”
59 why a comma after But?
106, “Consequently to extraction, library preparation was the subsequent step”
153-158 run on sentence, somewhat confusing ???
179 “baptized”??? Was a priest present when this deletion was detected?
306 – should be “understudied”?
Then, the biggest issues seem to be with understanding of biological or analysis aspects:
1. “the S protein expressed on the surface of the virion” is wrong. Expression occurs in cells, not in virions, as the authors may learn from a biology textbook.
2. RT-PCR being “the gold standard for the diagnosis of Covid-19 in behalf of its sensitivity, thereby many industries have developed commercial kits to reduce the typical run time and improve the methodology protocol” – it was perhaps OK to say that in 2020, at the peak of the panic, but in 2023 this indicates the authors not being up to date. They probably can learn from newspapers (or scientific literature) that most PCR tests are no longer enforced or even considered throughout the world…
3. “assembled on the Wuhan reference genome” – the right word is “aligned”
4. The link provided in reference 17 for SOPHIiA DDM software leads to an advertising page. It appears that there is no way to use the software without signing up and maybe making a payment, but some details should be given as to how the site was used. OTHERWISE, IT WOULD BE PRUDENT TO SEE THIS AS A POTENTIAL CONFLICT OF INTERESTS and reject the paper.
5. Equally important, the authors need to specify how the Nextclade site was used. Some details should be provided as to how the site in reference 19 was used. Was the command tool or Web application used? What arguments, parameters, or settings were used?
6. “threshold set for the calling of mutations (Variant Fraction, VF) was 70% of the entire number of mapped reads, hence a specific alteration has been annotated was present above.” Is really poorly worded. I assume they mean 70% of reads covering a particular coordinate, otherwise it makes no sense whatsoever. It‘s not justified in any way. Given the huge excess of the N gene sgmRNAs, one could argue that a mutation in sgmRNA template might lead to 70% of mutated N gene sgmRNA reads but not reflect a genomic change. Such cases need to be investigated by manual inspection and IGV screeshots would be helpful.
7. Even more disturbing, ref 22 returns “File not Found. Sorry, the content you are looking has disappeared.”
8. “probabilistic models were generated associated to an Angstrom error estimate” – what?
9. “The entire extension of the N gene was investigated” – what extension is that?
10. Fig 1 x-axis is unreadable.
11. The authors lis some mutations without any corroborating evidence. They should have shown IGV figures for each of these deletions, say, in the Supplement. As such I don’t believe their claims – there are too millions of many poorly constructed genomes in GISAID already.
12. They have the raw data – they should deposit that in SRA. Then others could actually analyze these results properly.
13. The whole “3D structure analysis” sounds pretty meaningless. Has the function of the N changed?
14. A database with “over 8.000 sequences” is mentioned but is not available in any form or shape. Mentioning that is like saying “my cousin in Milan has seen some genome deletions but I can’t show them”. And in English-speaking journals, 8.000 means 8.
Conclusion: the paper very poorly written, it is not biologically sound, methods are poorly presented, results are not convincing, datasets not made public.
Reviewer 3 Report
The manuscript analyzed N gene sequences of samples from Alpha, Delta, and Omicron variants and described observed deletions. The aim of the study and the interpretation of the results is unclear. The title and abstract indicate that deletions in N genes can be spuriously identified due to technical issues in qRT-PCR. However, the supporting evidence is unclear. The manuscript requires extensive improvement.
1. The results of spurious deletions due to RT-PCR errors need to be clearly presented.
2. The manuscript needs to clearly describe how spurious deletions were distinguished from real deletions. Intuitively, I think it is impossible. From the comparison between sample sequences (section 3.1), it is impossible to distinguish spurious from real mutations. Similarly, analysis of mutation frequencies (section 3.3) cannot tell which ones are spurious. Low-frequency mutations can be simply rare mutations. To conclude spurious mutations, other evidence is necessary.
3. Figures: Please use a bigger font size.
4. Line 174-175: I could not find the different mutation of three nucleotides in 7 Alpha and 2 Omicron samples in Figure 1. Please highlight them in the figure. Similarly, I could not the sample number 11 without the previous mutations and its deletion (line 178). Overall, in Figure 1, it is difficult to find the mutations described in Lines 174-201. Please highlight the mutations in Figure 1.
5. A figure presenting the local consensus sequences (LCSs) with detected mutations will be useful to follow the description of the characteristics of LCSs (Section 3.2 LCS comparison).